# Discovering Cellular Mitochondrial Heteroplasmy Heterogeneity with Single Cell RNA and ATAC Sequencing

**DOI:** 10.3390/biology10060503

**Published:** 2021-06-05

**Authors:** Aidan S. Marshall, Nick S. Jones

**Affiliations:** Department of Mathematics, Imperial College London, Huxley Building, South Kensington Campus, London SW7 2AZ, UK; a.marshall19@imperial.ac.uk

**Keywords:** scRNA-seq, scATAC-seq, mitochondria, heteroplasmy, heterogeneity

## Abstract

**Simple Summary:**

Mitochondria, the powerhouse of the cell, exist in the range of 100 s–1000 s of copies in almost every cell in the body, each with their own mitochondrial DNA, called mtDNA. When the healthy operation of a significant proportion of these mitochondria is disrupted, it can lead to dysfunction and by extension disease. One source of dysfunction arises due to mutations in the mtDNA, resulting in individual cells harbouring multiple versions of mtDNA—a “standard” wild type and a variant—a state called heteroplasmy. Heteroplasmy is a state that can arise either through inheritance or by mutations that occur through life, resulting in a new mitochondrial allele within a cell. The proportion of mitochondria that have a wild type and that have a variant allele differs between individuals, tissues within an individual, and even cells within a tissue. Historically, heteroplasmy has mainly been studied with bulk sequencing technologies, which miss variation within a tissue. The cellular variation in heteroplasmy throughout the body and its implications for pathology is not fully understood. In this review article we outline recent developments in *scRNA-seq* and *scATAC-seq* techniques which allow researchers to discover the extent of this cellular variation and further uncover the role heteroplasmy plays in disease at the cellular level.

**Abstract:**

Next-generation sequencing technologies have revolutionised the study of biological systems by enabling the examination of a broad range of tissues. Its application to single-cell genomics has generated a dynamic and evolving field with a vast amount of research highlighting heterogeneity in transcriptional, genetic and epigenomic state between cells. However, compared to these aspects of cellular heterogeneity, relatively little has been gleaned from single-cell datasets regarding cellular mitochondrial heterogeneity. Single-cell sequencing techniques can provide coverage of the mitochondrial genome which allows researchers to probe heteroplasmies at the level of the single cell, and observe interactions with cellular function. In this review, we give an overview of two popular single-cell modalities—single-cell RNA sequencing and single-cell ATAC sequencing—whose throughput and widespread usage offers researchers the chance to probe heteroplasmy combined with cell state in detailed resolution across thousands of cells. After summarising these technologies in the context of mitochondrial research, we give an overview of recent methods which have used these approaches for discovering mitochondrial heterogeneity. We conclude by highlighting current limitations of these approaches and open problems for future consideration.

## 1. Introduction

Mitochondria play a central role in most human cell types, and by extension, ensuring their healthy function is critical. When such mitochondrial function is perturbed, substantial pathology can ensue, resulting in debilitating mitochondrial diseases [1]. Unlike other organelles found in animal cells, mitochondria possess their own DNA, referred to as *mtDNA*, which exists as 100 s to 1000 s of copies per cell. Individual humans universally possess a mixture of mitochondrial genomes, a state referred to as *heteroplasmy* [2]. The precise allele fraction of this mixture at any given locus can vary from no heteroplasmy, 0%, all the way up to 100%, termed *homoplasmy*, in which a mutant variant has taken over the population of mtDNA. The state of heteroplasmy can originate through inheritance down the maternal line or via somatic mutations that occur during development or later in life. Originally, mitochondrial mutations were hypothesised to have arisen due the exposure of mtDNA to large amounts of reactive oxygen species. However, the pattern of mutations is more consistent with errors that take place during replication [3]. Numerous diseases, as well as aspects of ageing, have been linked with mitochondrial mutations and the extent to which these mutations are pathological can greatly depend on their heteroplasmy [4,5].

The level of heteroplasmy is heterogeneous at the population, individual, tissue, and cellular level [6,7]. Whilst some studies focus on specific mitochondrial mutations which are pathological or have been associated with mitochondrial diseases [8,9], the focus of the current review is on methods which discover mutations with high throughput to uncover deeper heterogeneity. Next-generation sequencing (NGS) has been a key technology for investigating many aspects of heteroplasmy. However, many early studies were conducted using bulk sequencing techniques, which entail homogenising cells from a tissue, placing an inherent limit on the resolution of these approaches and preventing analysis of cellular variation [2,10]. In recent years, single-cell omics approaches have revolutionised biological research, enabling levels of granularity that were previously not possible, with several recent studies making use of these technologies to study the heterogeneity of heteroplasmy within a tissue. To give one example, lineage tracing is the process of identifying a cell’s progeny and grouping the descendent cells together through the use of a labelling strategy. Lineage tracing has applications in fields including stem cell biology for understanding cell fate decisions, and in cancer biology to infer how tumour heterogeneity arises with implications for treatment resistance [11]. A body of recent work has demonstrated that mitochondrial mutations are projected down cell lineages, and that this can be observed using single-cell data, making the mitochondrial genome an effective in vivo barcode. Beyond lineage tracing, researchers should in principle be able to obtain vital mitochondrial features at the single cell level: both the presence of mutations, and the heteroplasmy of those mutations. Single-cell approaches are further advantageous in that observed mutations can be cross-validated in other cells to protect against sequencing error. However, they also offer the prospect of detecting mutations which would be unobservable in homogenised mixtures, which would be expected of some somatic mutations [12]. Together with features of cell state, this data would facilitate research in understanding the mechanisms by which the accumulation of heteroplasmy leads to pathology.

In this review, we begin by reporting what methodological progress has been made in exploring aspects of the mitochondrial genome in single cell data thus far, what its limitations are, and what the future scope is for using this data to further explore heterogeneity in heteroplasmy at the single cell level. A particular emphasis will be placed on scRNA-seq and scATAC-seq, two of the most popular single-cell techniques currently in use with the capability of obtaining information about cell state as well as mitochondrial genotype. Though we provide a succinct overview of scRNA-seq and scATAC-seq to contextualise the discussion with regards to mitochondrial DNA, we direct readers to the numerous excellent reviews covering the experimental and computational methods associated with these protocols [13,14,15,16].

## 2. Background on Single-Cell Omics

Single-cell data is omics data derived from individual cells [17]. These techniques encompass single-cell RNA sequencing, epigenomics, immune profiling, copy number variation, and proteomics, among others. Additionally these techniques can be combined to obtain two or more sets of features from the same cells, referred to as *multi-omics* [18]. Prior to the development of high-throughput single-cell techniques, most studies were conducted in bulk analyses, where numerous cells were homogenised and processed at once. Whilst such analyses have yielded useful insights [10], they are inherently unable to detect heterogeneity at the single cell level meaning cellular states relevant to pathology went unobserved. The breakthrough of single-cell RNA sequencing changed this. The original inception of scRNA-seq was confined to studying five cells, though recent developments have greatly expanded throughput by several orders of magnitude [19,20]. This has been enabled through the use of innovations which simplified and automated isolation of cells, as well as intelligent strategies for barcoding individual RNA molecules from different cells, enabling multiplexed sequencing of RNA from numerous cells at once [21]. This development has resulted in massively parallel, commercially available techniques such as those offered by 10× genomics, which can allow for >100,000 s of cells to have their transcriptomes sequenced simultaneously in a single experiment [20]. Other single-cell omics are following suit with developments in obtaining high-throughput [22]. By leveraging these recent technological breakthroughs, large consortia such as the *Human Cell Atlas* and *LifeTime* aim to use sequences from 1,000,000 s of cells to further understand human pathology at the single cell level [23,24].

The granularity combined with the scale of these techniques has had a major impact an many areas of biology, such as immunology where scRNA-seq has been used to help define cellular states, observe heterogeneity in genomic sequences of cells derived from the same donor, and single-cell immune profiling can be used to measure TCR sequences to perform lineage tracing and measure TCR repertoires [25]. A similar impact has been made in tumour research where single-cell techniques have been used to probe heterogeneity throughout the tumour micro environment, and to perform lineage tracing of sub-clonal populations within the tumour [26,27]. In contrast to these areas of biology, the use of single-cell omics data to probe single-cell mitochondrial heterogeneity and how it interacts with cellular phenotype remains relatively underdeveloped, even though they can provide coverage of the mitochondrial genome [28].

## 3. Sequencing Protocols & mtDNA

For researchers interested in discovering heterogeneity in mitochondrial heteroplasmy with single-cell data, two primary tasks must be performed—(1) identify real mitochondrial mutations and (2) accurately assess the heteroplasmy of these mutations. This is achievable as single-cell sequencing data obtains sequences derived from the mitochondrial genome in individual cells. We focus on two popular modalities for obtaining such sequences, scRNA-seq and scATAC-seq, due to their wide availability and because of their potential to combine observations of cell state with mitochondrial genotype. Though RNA does not measure heteroplasmy directly, heteroplasmy in DNA should broadly be reflected in RNA sequences. Additionally, RNA modifications in mitochondrial genes could be another axis of variation which could be phenotypically relevant within single cells [29]. The primary purpose of these two distinct approaches is to quantify gene expression and find accessible regions of the genome to quantify DNA regulation respectively. RNA sequences from scRNA-seq and DNA sequences for scATAC-seq enable the discovery of mitochondrial heteroplasmy heterogeneity by finding mismatches in these sequences from the reference genome, termed *variant discovery*.

A challenge posed by single-cell data, however, is the variation in quality metrics relevant to achieving these two goals. Due to inconsistent terminology surrounding quality metrics in the literature, we clarify and standardise our use of language surrounding the metrics used throughout this review. As is described in more detail below, sequence data comes in the form of many separate sequences termed as a *library* with each of the constituent sequences referred to as a *read*. Within a library, any given locus on the genome can typically be represented multiple times in separate reads. Having more sequence reads at any given position on the genome is desirable as it makes mutation detection more reliable by protecting against sequencing errors and enables more accurate estimation of the heteroplasmy of a mutation at a given position. Building on the clear terminology from Sims et al. (2014) [30], we use the following definitions: *Depth of coverage* is a property of a single site on the genome and is the number of times a base in the reference genome is covered by a high-quality read; *Breadth of coverage* is a property of a target sequence on the genome and is the proportion of bases on the target sequenced to a desired depth of coverage. The primary target genome is the mtDNA in this article. Moreover, as we are interested in cellular heterogeneity, we are interested in these quantities *per cell*. Both quantities also depend on parameters such as sequencing base quality thresholds, in which some positions from a read are neglected due to low quality sequencing.

Figure 1 illustrates these definitions with data from two contrasting scRNA-seq data types, showing the breadth and depth of coverage of the mt-ND1 gene in two human data sets. Using mt-ND1 as the sequence of interest, Figure 1a figure shows the number of reads aligned to each position in the gene. 10× data (blue) shows high mean depth of coverage per cell for positions >4000. For positions less than this, 10× data shows shallow depth of coverage, and can therefore be described as having narrow breadth of coverage restricted to the 3${}^{\prime }$ end of the transcript. Smart-Seq2 data (red) in contrast to this has superior mean depth of coverage across almost the entire length of mt-ND1 and can therefore be said to have superior breadth of coverage to 10× sequencing data. The intervals around the mean emphasise that these definitions are made per cell, with considerable variation in both depth and breadth of coverage which can be attributable to biological signal, such as transcriptional variation, as well as technical stochasticity in detection.

Figure 1b shows how the breadth of coverage per cell declines across the whole mitochondrial genome as the demand for depth of coverage increases. Almost every base in the mitochondrial genome is covered by at least a low number of reads in some cells. 10× data shows a rapid decline in the breadth of coverage as the depth threshold is increased, whilst Smart-seq2 data shows greater stability.

### 3.1. scRNA-seq

RNA sequencing has been around in excess of a decade, and is now ubiquitous in biological research. In this technique, RNA is extracted from a biological tissue, complementary cDNA is synthesised by reverse transcription, then PCR amplified, resulting in what is referred to as a *library of cDNA reads*. This library is then sequenced on a high-throughput sequencer. Many variations of this basic procedure exist, and more have arisen since the emergence of *single cell* RNA sequencing (scRNA-seq). Instead of aggregating and homogenizing many cells from a tissue together, scRNA-seq is capable of capturing, amplifying, sequencing, and then quantifying reads of individual cells separately, allowing for the the expression of each gene to be quantified on a per cell basis. The transcript counts per cell for each cell in a sample is typically then compiled into an array of data referred to as the *expression matrix* which is then used for downstream analysis [16].

scRNA seq protocols can be divided into two distinct categories, *full-length transcriptomics*, and *3${}^{\prime }$/5${}^{\prime }$ transcriptomics* approaches based on what length of the RNA transcript is captured and sequenced, each with its own advantages and drawbacks.

#### 3.1.1. Full-Length Transcriptomics

Full-length protocols sequence the entire length of a transcript. There are two typical approaches to full-length transcriptomics. One method, followed by popular approaches such as the Smart-Seq2 protocol [33], is to fragment a library of full-length cDNAs into shorter molecules and followed by short-read sequencing on a short-read Illumina sequencer. Another approach is to use long-read sequencing on a platform such as Oxford Nanopore or Pacific Biosciences. Long-read-length sequencing is advantageous in that the full sequence of a gene’s RNA is obtained, which is particularly useful for detecting variants across the length of a gene. The primary drawback of direct long-read protocols is that they are low throughput and costly to perform [13]. This is because cells are typically placed into separate wells on a plate and sequenced separately, bounding their throughput. Additionally, the long-read length sequencers offered by Oxford Nanopore or Pacific Biosciences suffer from higher error rates than short-read Illumina sequencers, posing problems for analysis at the single nucleotide level as would be done to investigate mitochondrial heteroplasmy [34]. The ability to sequence the entire length of transcripts, however, is attractive as it has the potential for discovering variants across most of the mitochondrial genome at a cellular level.

#### 3.1.2. 3${}^{\prime }$ and 5${}^{\prime }$ Droplet-Based Transcriptomics

Alternatives to standard full-length approaches are 3${}^{\prime }$ and 5${}^{\prime }$ methods in which reads are reverse transcribed from the 3${}^{\prime }$ or 5${}^{\prime }$ end of a transcript. These approaches do not make use of fragmentation in the same manner as Smart-seq2, thereby resulting in only one cDNA molecule per transcript prior to PCR. In contrast to full-length approaches, 3${}^{\prime }$ and 5${}^{\prime }$ protocols have been able to achieve immense throughput, and high accuracy, being sequenced on an Illumina sequencer. This has largely been achieved by microfluidics technologies which capture single cells in oil droplets containing cellular barcodes which are attached to each read. This barcoding protocol allows reads from many separate cells to be pooled together (referred to as *multiplexing*), PCR amplified and then sequenced, improving throughput. Reads can then be de-multiplexed by using software that exploits the pre-appended barcodes identifying each cell. In other words, intelligent barcoding strategies for cell identification enable massive scaling up of throughput. Such an approach is routinely used by three of the most popular scRNA sequencing platforms—inDrop, Drop-seq, and 10× chromium—and can be easily used to sequence 100,000 s of cells in a single experiment [20,35,36]. Additionally short-read technologies enable the use of *unique molecular identifiers*.

#### 3.1.3. Unique Molecular Identifiers

In protocols using unique molecular identifiers (UMIs), each indvidual cDNA read has a random nucleotide sequence attached during the reverse transcription process, with such random nucleotide sequences being referred to as UMIs. Each resulting cDNA molecule therefore has both a cell barcode and a unique molecular identifier attached. Attachment takes place during the first step of library preparation, and importantly, before PCR amplification. cDNA libraries undergo multiple rounds of PCR amplification, resulting in multiple copies of every original molecule from the cell, and has been shown to result in PCR bias, in which some molecules are amplified more than others [37]. This is significant for researchers who wish to estimate the proportion of sequence molecules (RNA or otherwise) carrying a certain variant as it is possible that variation induced by PCR could be a source of noise in the resulting values of variant allele fractions. This could be particularly relevant for mitochondrial researchers, for whom accurate heteroplasmy measurements are important. By allowing groups of PCR amplified reads to be grouped together based on their UMI, reads can be de-duplicated to enable robust estimation of the original number of transcripts from each gene, and is the basis of the output from tools like STARSolo and CellRanger which give a UMI counts based expression matrix to quantify gene expression [38]. Non-UMI approaches rely on read counts without any correction [39]. Though the use of UMIs to form robust counts has been addressed by software tools for sequence counting such as UMI-tools [40], the ability to form a consensus read sequence based on all the copies of a molecule remains nascent, with numerous computationally intensive approaches having been suggested, but few used broadly.

Such UMI approaches are typically limited to short-read sequencing, in which only positions within 100 s of bases of the 3${}^{\prime }$ or 5${}^{\prime }$ end of a transcript are covered, whilst full-length approaches have typically been missing UMIs. Smart-Seq2 like approaches prevent UMI usage due to the fragmentation step which would require linking every fragment to the source UMI, something which remains challenging. Direct full-length sequencing approaches using Oxford Nanopore or PacBio also inhibit this approach due to the high error rate of long-read sequencers (nearly 50% of UMIs can have at least one error in these approaches) which inhibits robust UMI grouping as sequence errors can create spurious UMIs which did not tag any original molecule. Additionally, long-read sequencers typically obtain fewer reads per UMI than short-read sequencers further compounding the issue of accurate UMI usage. Efforts are being made to address these obstacles [41,42,43]. Smart-seq3 has recently been developed as a full-length UMI protocol, though it has not yet attained widespread usage [44]. Such problems are not associated with short-read Illumina sequencers which have a low error rate enabling accurate barcode-UMI sequencing. Whilst this is the conventional approach to sequencing libraries prepared with 10× technologies [20], it does entail fragmenting prepared cDNA reads into two short sequences—one containing the barcode and UMI sequences, one containing the genomic sequence—of approximately 100–200 bps, resulting in substantially less breadth of coverage of the genome than a full-length approach.

#### 3.1.4. Comparing Full-Length and 3${}^{\prime }$/5${}^{\prime }$ for Mitochondrial Heterogeneity Discovery

We can understand the quality of these two approaches for discovering mitochondrial heterogeneity by using data from Smart-Seq2 and 10× Chromium platforms as exemplars [31,32]. Figure 2 shows the depth of coverage across the mitochondrial genome on a per cell level for each of these approaches. Smart-Seq2 displays consistently high depth of coverage across the mitochondrial genome. In comparison, 10× data shows far lower breadth of coverage with consistent enhancements in depth of coverage near the 3${}^{\prime }$ end of each gene. Whilst this non-uniform depth of coverage might prevent identification of variants across the mitochondrial genome variants should be consistently identifiable within close proximity to the 3${}^{\prime }$ end.

We summarise the trade-off between popular full-length sequencing protocols and those which use 3${}^{\prime }$/5${}^{\prime }$ droplet-based approaches as follows: full-length approaches typically maintain a superior breadth of coverage, but lack UMIs and can have a higher sequencing error rate when used in conjunction with long-read sequencing. In contrast, 3${}^{\prime }$/5${}^{\prime }$ droplet-based approaches incorporate UMIs into their analysis, meaning that biases and errors arising from PCR can in principle be ameliorated, and sampled transcripts can be accurately counted. Cells sequenced with these 3${}^{\prime }$/5${}^{\prime }$ library approaches, however, show great heterogeneity in precisely which bases attain a sufficiently high depth of coverage for accurate variant calling in any given cell. Furthermore, the breadth of coverage of 3${}^{\prime }$/5${}^{\prime }$ droplet-based approaches is circumscribed, with high depth of coverage localised around the corresponding transcript end due to the end bias of these approaches, leaving a lot of the genome with insufficient depth of coverage to inspect heteroplasmy [45]. Comparing variants across different cells using such approaches should be possible, but requires care to account for coverage variation.

### 3.2. scATAC-seq

Single-Cell Assay for Transposase Accessible Chromatin using sequencing (scATAC-seq) is another popular technique capable of obtaining sequence data from individual cells. Briefly, scATAC profiling methods make use of a hyperactive transposase Tn5 which binds preferentially to regions of accessible chromatin with the simultaneous ligation of adapters to the ends of the associated DNA molecules which can then be PCR amplified [15]. Regions of the genome which have accessible chromatin will then typically tend to have greater coverage than other areas of the genome. These differences in coverage can then be used to perform *peak calling* in which regions which have a far larger number of aligned reads than background are then used to infer the presence of accessible regions [15]. These accessible regions are associated with cis-regulatory elements which play a key role in transcriptional regulation, which scATAC-seq therefore measures.

The use of cellular barcoding strategies has enabled the development of high-throughput droplet methodologies similar to those used for RNA sequencing, and there are also commercially available platforms such as those offered by BioRad and 10× Genomics [22,46]. Unlike scRNA-approaches, UMI methodologies are not as mature and are not as widely used. scATAC UMI-based approaches are, however, being developed to mitigate amplification biases. Standard approaches de-duplicate reads based on the start positions of where reads align to the genome are established [47]. However, this can induce further biases. By making use of UMIs, one study was able to quantify how many Tn5 insertions may independently occur at the same position [48]. This showed that up to 20% of the reads discarded based on their alignment positions, a standard method step in many protocols, were done so erroneously and actually arose due to independent Tn5 insertion events. As de-duplication alters the counts of reads which carry variants or not, this approach could bias heteroplasmy estimates.

In early ATAC-seq approaches mitochondrial reads accounted for an overwhelming proportion of aligned reads. This is because nuclear DNA regions only come in 2 copies in any cell, whilst mtDNA comes in 1000 s of copies, and each mtDNA is accessible as they are not bound up in a nucleosome. Consequently, a lot of the signal from the nuclear DNA was considered to be drowned by the “noise” of mtDNA reads, and efforts were taken to suppress the coverage of the mtDNA in these sequencing approaches. More recent high-throughput single-cell protocols for ATAC sequencing typically have reduced mitochondrial content as these protocols isolate individual nuclei, discarding the rest of the cell [22]. Despite this, these approaches still offer the potential to probe mitochondrial heterogeneity as mitochondria have been closely associated with the peri-nuclear sheath, meaning total isolation is unlikely to occur [49].

## 4. Heteroplasmy in Single-Cell Data

In this section we review previous work that has been undertaken to probe mitochondrial heteroplasmy in single-cell data. This centres on the work of three previous papers, one that analysed conventional scRNA sequencing protocols to perform lineage tracing, and two that developed new protocols to gain greater insight into the mitochondrial variants at the single cell level.

Whilst some studies have investigated mitochondria in single cells their throughput has been limited, e.g., [50]. Others studies seek to inspect mutants known to be pathogenic, or have required producing clonal populations of cell’s [8,51]. The focus of the current article is on approaches which offer both high throughput and scope for discovering previously unobserved heteroplasmy from in vivo samples. Each of the following papers focuses on using mitochondria for lineage tracing, and all demonstrate how mitochondrial heterogeneity in both the mutations found and their range of heteroplasmies can be discovered using these techniques.

### 4.1. Mitochondrial Heterogeneity with scRNA-seq and scATAC-seq

In vivo lineage tracing using single cells conventionally relied upon the detection of nuclear somatic variants. By performing variant calling in scRNA-seq and scATAC-seq data, Ludwig et al. (2019) showed that mitochondrial sequences could in certain circumstances be used as natural genetic barcodes for tracing cellular lineages [52]. The authors argue that mitochondrial sequences are short enough to allow for cost-effective sequencing, yet are long enough to harbour numerous mutations. As such, mitochondrial sequences offer enough genetic diversity to serve as an effective barcode that can differentiate between cellular lineages. Grouping cells by the mitochondrial mutations that are shared between them should enable the inference of lineage.

They compare the percentage (i.e., breadth of coverage) of the mitochondrial genome, attaining different levels of depth of coverage for a range of single-cell RNA sequencing techniques. Short-read, 3${}^{\prime }$ approaches show considerably less of the mitochondrial genome being sequenced at any given level of depth of coverage, as would be intuitively expected (see Section 3.1.4). Smart-seq2 has the greatest breadth of coverage at any fixed level of depth compared to the other approaches they analysed. By using a scATAC-seq derived from whole cells, rather than nuclei, using an earlier approach [53] they show that this modality gives uniform depth and breadth of read coverage across the mitochondrial genome, making it a suitable candidate for obtaining reliable heteroplasmy data from single-cell data.

Interestingly, the authors also present a comparison between the heteroplasmies observed in joint WGS and scRNA seq data from single cells. For the variants found in both datasets there appears to be a high level of agreement, demonstrating the utility of scRNA sequencing approaches to indirectly discover mutations in the mtDNA as well as their heteroplasmy. However, there are some variants which appear to be specific to the RNA modality. The authors claim and that some of these are attributable to RNA editing, and transcription errors. However, some of the low-heteroplasmy RNA variants are likely attributable to errors in RNA sequencing arising from some combination of reverse transcription errors—a step which is absent for WGS data—and errors occurring during early PCR, highlighting sources of false positive variants in scRNA data. High-heteroplasmy RNA variants are unlikely to be due to these systemic errors as the chance of reverse transcription errors, PCR artifacts, or sequencing errors independently producing errors at the same position in numerous reads, as would be required for an observed high heteroplasmy, is unlikely to occur by chance. High heteroplasmy variants are therefore more robust to multiple sources of error.

In vitro, the authors performed variant calling across cells from 64 different erythroid and myeloid colonies, profiling between 8 and 16 cells in each colony. The donor and colony of origin was known, enabling a supervised analysis to be performed. Variants observed at near homoplasmy were capable of separating cells into their donor of origin as well as singling out individual colonies. It was observed that unique clonal mutations were found in many of the colonies, at a range of heteroplasmies capable of differentiating many of the colonies. They made similar observations using an scATAC-seq dataset published by Buenrostro et al. [53]. Both insights demonstrate how heteroplasmic mutations are propagated through colonies. Using publicly available T-cell, Smart-Seq2 [33] scRNA-seq data and TCR-seq data [54,55], Ludwig et al. were also able to demonstrate similar findings in vivo. Grouping T-cells by their TCR receptor allowed for a similar supervised analysis to take place, in which mitochondrial mutations were shared among cells sharing TCR sequences. Some mtDNA mutations were capable of further refining the clusters identified by identifying sub-populations within TCR groups, whilst some mtDNA mutations were shared across TCR groups, suggesting a common ancestor before V(D)J recombination took place.

In addition, by using somatic mitochondrial mutations first identified in bulk RNA-seq Ludwig et al. further elucidate clustering of cells derived from colorectal adenocarcinoma primary tumors when using scRNA-seq to further resolve which cells did and did not possess these mutations, highlighting the ability of mtDNA mutations to elaborate intratumoral clonal heterogeneity. By using publicly available chronic myelogenous leukemia scRNA-seq dataset [56], tsne clusters based on mitochondrial genotypes as features showed nearly perfect separation of cells into their originating donors. Finally, analysis was performed of 10× chromium 3${}^{\prime }$ using the PBMC data in the original 10× chromium publication [20]. These PBMCs were taken from a recipient before and after transplantation of HSCs. Two homoplasmic mutations were observed to distinguish the donor and recipient, and post transplant PBMCs from the recipient were found to be overwhelmingly derived from the donor, demonstrating the potential of 3${}^{\prime }$ approaches to detect heterogeneity in spite of their narrow breadth of coverage. These observations highlight the potential for finding biological insight regarding heterogeneity in heteroplasmy at the cellular level through the use of single-cell data.

### 4.2. EMBLEM

Epigenome and Mitochondrial Barcode of Lineage from Endogenous Mutations (EMBLEM), which focuses on single-cell ATAC sequencing to perform lineage tracing using mitochondrial variants, was a strategy presented by Xu et al. [57]. Just as lineage tracing in scRNA seq allows users to get cellular lineage information and cell state information from the same cells, EMBLEM aims to let users obtain epigenomic state alongside clonal genotypes.

As noted by the authors, observing a mutation at x% heteroplasmy in bulk could arise due to x% of cells harbouring that mutation at homoplasmy or all cells separately harbouring that mutation at x%, as well as other combinations between these extremes. Inspecting the variant allele fraction of different mutations in single cells using scATAC-seq, Xu et al. saw that different mutations occupying different positions along this spectrum with some mutations being present in many cells at low heteroplasmy, whilst other mutations are found at high heteroplasmies in single cells.

By sequencing LSC and blast cell populations Xu et al. showed that they shared mitochondrial mutations at similar variant allele fractions. This is coherent with the established hierarchical lineages in which LSCs give rise to blast cells and should therefore be expected to share mitochondrial mutations [58]. Taking a sample from a single patient they found several cells that shared the same four mutations, with subsets of these mutations contained in many other cells. Combining this observation with the fact that all cells are derived from the same progenitor cell and should therefore share the same mutations, the authors aimed to quantify the proportion of cells a mutation would be observed in as function of mitochondrial read depth. Mutations with moderate ∼20% heteroplasmies could be detected in ∼90% of cells with only 20 reads covering the position of that mutation. Conversely, when a mutation has heteroplasmy <1%, more than 100 read depth can be required to obtain a detection rate exceeding 90%, reflecting the need for greater sampling to detect low heteroplasmy variants. Moreover, without deep sampling, such low-heteroplasmy mutations would be challenging to distinguish from sequencing error.

These observations highlight the level of confidence which mitochondrial variants can be called with accuracy, showing that near homoplasmic variants can be called robustly and discovered even when read depth is low at a position. This has implications for detecting mtDNA mutations in other sequencing types, such as 10× 3${}^{\prime }$, where depth of coverage at different bases of the mitochondrial genome is heterogeneous both between cells and across the genome due to the variability of 10× coverage at the cellular level (see Figure 2b). In such sequencing types, high-heteroplasmy mutations should be observable in regions with low depth of coverage away from transcript ends, whilst regions of the genome with high depth of coverage should be able to discover mutations with low heteroplasmy.

### 4.3. mtscATAC-seq

Building on the earlier work by Ludwig et al. (2019) [52], a new scATAC-seq protocol was developed by Lareau et al. (2021) to perform mitochondrial genotyping in single cells [59]. Ludwig et al. (2019) showed the limited breadth of coverage of the mitochondrial genome in high-throughput 3${}^{\prime }$ sequencing assays such as Drop-seq. They show similar shortcomings for the massively parallel and massively popular 10× chromium system when compared to a full-length, low-throughput assay like Smart-Seq2 [52]. In order to capture the best of both of these approaches, that is to say, a high-throughput approach with uniform coverage of the mitochondrial genome, Lareau developed mitochondrial single-cell assay for transposase-accessible chromatin with sequencing (mtscATAC-seq). As mentioned previously, most scATAC protocols have been developed to work with single nuclei, which depletes the mitochondrial coverage. By modifying scATAC-seq, the authors develop an assay that works with whole cells.

This protocol is a modification of the scATAC-seq system offered by 10× genomics. Usually this uses pooled nuclei before adding the Tn5 enzyme. Instead of this, Lareau et al. use “mild lysis” of whole cells in combination with fixation. This enables cells to take up the Tn5 enzyme whilst preventing the leakage and cross contamination of mitochondria between between cells in suspension. After masking NUMTs (see Section 5.6 below) they find that their approach shows remarkable uniformity across the mitochondrial genome, with residual coverage variation reflecting stochasticity in PCR and Tn5 insertion variability. They show that their approach leads to an approximately 20-fold increase in the mean depth of coverage of the mtDNA genome compared to the original protocol achieving an mean read depth of 191 across the mitochondrial genome, across cells. In general, their method aimed to have approximately 20 reads after PCR duplicate removal.

One of the key features of this work is the development of Mitochondrial Genome Analysis Toolkit (mgatk). In contrast to some variant callers which have aimed at genotyping single cells, mgatk aims to analyse clonal mutations. Taking the .bam file output fromCellRanger, mgatk produces a matrix of de-duplicated per-cell, per-strand count of all alleles at all positions in the genome. Here, de-duplication groups all reads sharing a starting position and the read with the highest mean base quality is then used as a representative read. This de-duplication procedure should enhance heteroplasmy estimation by counteracting PCR amplification biases. ATAC-seq obtains reads from both the forward and reverse DNA strands. For each mutation, mgatk measures the correlation between the allele counts on both strands across the dataset of cells and is termed *strand concordance* which helps to protect against photobleaching effects (see Section 5.2). Only mutations with a high strand concordance and variance mean ratio in their heteroplasmies are retained for further analysis. mgatk then counts all cells where the variant was detected on both strands in at least two reads. Lineage variants are considered as high confidence for lineage tracing if the variant is detected in five or more cells.

Using this variant calling approach, Lareau et al. were able to observe mitochondrial heterogeneity in a range of phenomena including identification of clonal subgroups using mitochondrial mutations in TF1 cells, refine clonal substructure in PBMCs and measure an association between cell’s chromatin profiles and their clonal sub-grouping, thereby linking genotype and phenotype. Additionally, observations were made of the level of heteroplasmy ranging from 0 to 100% in the 8344A>G mitochondrial mutation among cells from an individual suffering from the chronic mitochondrial disease, myclonic epilepsy with red ragged fibers (MERRF) [60,61], which this mutation is associated with. In adjacent work by Walker et al. (2020), the mtscATAC and mgatk protocol was applied to PBMCs from 3 patients suffering from mitochondrial encephalomyopathy with lactic acidosis and stroke-like episodes (MELAS), commonly associated with the A3243G mutation [9]. Of all the cell types present, T-cells consistently had this mutation at lowest heteroplasmy across all 3 donors, showing how single-cell approaches can discover mitochondrial heterogeneity among different cell types. These examples highlight the potential of high-throughput approaches to discover mitochondrial heterogeneity.

### 4.4. MAESTER

A recent pre-print by Miller et al. [62] develops a high-throughput scRNA-seq approach. As previously discussed, Smart-seq2 is able to offer far superior breadth of coverage to the 10× 3${}^{\prime }$ platforms, making it superior for discovering variants that can be used for lineage tracing, yet is limited by its throughput. The authors present Mitochondrial Alteration Enrichment from Single-cell Transcriptomes to Establish Relatedness (MAESTER), which seeks to address this by enriching full-length mitochondrial cDNA which is produced in an intermediate step by platforms like 10× but are usually only sequenced to partial length. Complementing this is the Mitochondrial Alteration Enrichment and Genome Analysis Toolkit (maegatk), an extension of mgatk. This exploits UMIs more extensively than previous approaches. By grouping all reads sharing the same UMI together, maegatk creates a consensus read by using the most common nucleotide at every position in conjunction with the base’s quality. This should ultimately protect against sequencing and PCR errors as well enhancing the accuracy of heteroplasmy estimates.

Though early in its development, MAESTER offers the exciting potential to discover mitochondrial variants across a high proportion of the mitochondiral genome, along with accurate heteroplasmy estimates combined with transcriptional cell state.

## 5. Future Directions and Open Problems in Assessing Heteroplasmy at the Single Cell Level

Having explored recent developments using single cells to explore mitochondrial heterogeneity, we now provide a survey of challenges, limitations, and open problems in these approaches.

### 5.1. PCR and Heteroplasmy

As we have noted throughout this review, heteroplasmy at the single cell level is a key axis of variation, with potential implications for cell state. As such, obtaining accurate estimations of heteroplasmy is important to be able to fully understand this. A simple approach to quantify heteroplasmy for each cell is to go through each position in the mitochondrial genome, count the number of reads which display an alternative allele at that position, and divide this by the total number of reads at that position [52,63]. This simple approach has proved effective. However, as sequencing reads are the result of PCR, the heteroplasmy of input biological molecules could be distorted by variation in the degree of amplification of different input molecules.

Approaches to address this amplification bias have used de-duplication of reads, in which all PCR duplicates are grouped together and one read is picked to represent those reads. mgatk does this by using GATK’s picard tools [47] to group reads by their starting position, as it is thought that reads derived from different biological molecules are unlikely to share a starting position by chance. Among reads sharing this starting position, the one with the highest mean base quality is chosen as a representative. Recent work has, however, shown that different reads all derived from the same starting biological molecule may not share the identical starting position due to *PCR stutter* (see Sena et al. (2018) for more details [64]). Grouping reads by their starting position could therefore also bias heteroplasmy quantification.

Building on mgatk, a recent pre-print by Miller et al. (2021) developed the Mitochondrial Alteration Enrichment and Genome Analysis Toolkit (maegatk) [62]. This toolkit is designed to work with UMI data by grouping all reads sharing the same UMI together and forming a consensus call at every position along the transcript. This approach should both mitigate sequencing errors and help identify accurately the heteroplasmic proportions of input cells.

### 5.2. Photobleaching and Strand Concordance

Intuitively, mitochondrial mutations are present on both DNA strands and should be present at comparable levels in DNA-based sequencing data. Lareau et al. suggest that strand concordance breaks down due to a technical confound which recurrently generates sequencing errors in certain regions of the mtDNA genome on one DNA strand due to the effect of surrounding G’s on successive cycles, a form of sequencing bias [59,65]. The complementary strand would instead having surrounding Cs. This bias is therefore reflected and corrected by observing strand concordance, a measure of agreement between the heteroplasmy observed on both strands of a sequence across cells in a dataset. Mutations with concordance below the threshold are then discarded to protect against the photobleaching effect.

As a default, mgatk uses these statistics of variants found across multiple cells to identify likely sub-clonal variants. Lareau et al. note that this approach is valid not only for mtscATAC, but also full-length scRNA-seq approaches such as Smart-Seq2 in which the strandedness of RNA transcripts is lost during double stranded cDNA synthesis [33]. Therefore observing stand concordance should still counteract photobleaching in these approaches. In reanalysing some of the data from their previous study [52], they in fact show that one variant which they had previously identified actually demonstrated strand discordance, only being present in one strand. However, they do show that a strong degree of overlap between the variants that they called using their old approach and with mgatk factoring in strand concordance and VMR, suggesting that variants may still be called with some degree of accuracy without factoring in strand concordance given other rigerous quality control thresholds.

As 3${}^{\prime }$ approaches retain the complementary strandedness of the original transcript molecule, strand concordance cannot be calculated, thus the authors state that mgatk is unsuitable for identifying high-quality lineage variants in this datatype. The software does, however, still offer the option of genotyping single cells obtained from droplet-based UMI sequencing approaches. In conjunction with suitable quality control thresholds, this approach could yield heterogeneity at the per cell level regardless of the ability to measure strand concordance in 3${}^{\prime }$ data. This underscores the technical considerations that have to be factored into answering any research question that may arise from variant calling scRNA-seq data.

### 5.3. Mitochondrial Cellular Transfer

It has been observed that mitochondrial transfer between cells is a common phenomenon, occurring in multiple contexts [66,67,68]. Such observations potentially undermine the use of, endogenous mitochondrial mutations as in vivo barcodes for cellular lineage tracing. Xu et al. applied EMBLEM to scATAC-seq data from a previous experiment which entailed the mixing of human and mouse T-cells [69]. They found that species-specific mtDNA and nuclear DNA always paired appropriately to the correct species. This, they argue, demonstrates that inter-cellular mtDNA transfer does not occur universally between cells. However, it remains unclear the extent to which non-observations of mitochondrial transfer between T-cells of different species generalises to eliminate mitochondrial transfer as a potential confounder for lineage tracing studies in other contexts.

Simulations by Ludwig et al. suggest that rates of intercellular transfer between cells would need to be high in order to for the transferred mutations to confound the analysis, as variant calls are typically done at a level to exclude low heteroplasmy variants [52]. Furthermore, the success of these lineage tracing approaches in a supervised context is also suggestive of the limited extent of mitochondrial transfer in some settings. However, more work needs to be done to conclusively eliminate mitochondrial transfer as a potential source of error in lineage tracing studies in other tissues.

### 5.4. Ambient RNA in Droplet-Based Approaches

Droplet-based approaches to single-cell data entail pooling cells together in a single suspension [20]. Cells are then separated into their separate droplets. Standard subsequent analyses assume that all reads in a droplet originate from a single cell. Common violations of this assumption are empty droplets, in which no cell is found and doublets in which two cells made it into the same droplet. Popular alignment tools such as STARSolo and CellRanger have approaches for identifying likely empty droplets and software is being developed to handle doublets [70,71]. A less well known violation of the standard assumption is *ambient RNA*, which is cell free RNA in the original cell suspension which can accompany individual cells into their droplets. Such ambient RNA could arise from other cells that were lysed or damaged within the suspension. Tools have been developed to correct the expression matrix in light of such observations [72,73]. Species mixing data suggests that 1–2% of observed transcripts within a cell could in fact be ambient RNA. This suggests a technical limit on attempts to elucidate clonal substructure of tissues by using low heteroplasmy variant calls, which could mistake ambient RNA shared in the suspension for a clonal variant.

### 5.5. Mitochondrial Gene Expression Proportion Quality Control

In the analysis of single-cell expression data, best practice recommendations include a quality control step to filter cells based upon the proportion of mitochondrial reads or trancsripts they have [16]. Both the tutorials for Seurat and scanpy, two popular single-cell analysis software packages, use a 5% threshold as a default [74,75]. This originated as a standard procedure for distinguishing low-quality cells from high-quality cells, where it was observed that broken cells with a loss of cytoplasmic content were observed to have a much higher proportion of mitochondrial reads than obviously intact cells [76]. Whilst claiming to be robust across cell types, such observations were predicated on T-cells, dendritic cells, and mouse embryonic stem cells. Bulk studies of the much more energy intensive tissue of the heart have found mitochondrial read proportions in excess of 30% of all reads, with considerable heterogeneity found across tissue types [77]. Moreover, a systematic study of single-cell data comparing mouse and human data suggests that different thresholds might be appropriate for different species and tissues [78]. Therefore researchers should take care to adjust their thresholds according to the subject of their study.

Whilst establishing reliable quality control thresholds might be challenging, they pose another problem for researchers interested in exploring the heterogeneity of heteroplasmy in single cells. Beyond lysis, another rationale for the removal of cells with high mitochondrial count proportions is the removal of potentially *stressed* or pre-apoptotic cells, as retaining such cells would result in clustering cells by their stress state, which researchers are commonly uninterested in [79]. As such, they are often removed from the analysis by default. However, mitochondrial mutations [80], as well as mitochondrial proliferation and copy number [81,82,83] have been associated with apoptosis. When paired with mitochondrial threshold effects [4] it is plausible that the default processing of scRNA-seq data, which removes highly stressed cells, might systematically bias the conclusions researchers draw about the impact of mitochondrial heterogeneity on cell state. The assumptions underlying such a quality control process are not fully explicated in standard tutorials, and the extent to which the standard approach to these analyses might be a source of confounding is not well understood.

### 5.6. NUMTs and Heteroplasmy

Nuclear sequences of mitochondrial origin (NUMT) are sequences found within the nuclear genome which have a homologous relationship and therefore high sequence similarity to sequences of the mitochondrial genome. NUMT sequences can be found in humans as well as other species [84,85]. Such sequence similarity poses a problem for short-read sequencing experiments in which aligners map reads to positions of the genome based on sequence similarity and can results in false positive and false negative variants being called. False negatives can arise due to mtDNA reads being mapped to the NUMT positions in the nuclear genome, and false positives can arise due to NUMT reads being mapped to the nuclear genome. Such effects have the potential to confound inferred variant allele frequencies, skewing the heteroplasmy estimates of single cells. Computational study has shown these misalignments result in erroneous loss of coverage for the mitochondrial genome for reads derived from the mitochondrial reference genome in silico, and more severe loss of coverage can result when reads harbour variants that differ from the reference sequence. Some variants are even capable of completely removing all alignment from the mitochondrial genome [86]. By causing mutated reads to align to homologs in the nuclear genome, a greater proportion of wild type reads align to the mitochondrial genome, thereby reducing the measured heteroplasmy. Low heteroplasmy variants may simply go unobserved as a result. For more details regarding this, we highly recommend Maude et al. (2019) [86].

The degree to which NUMTs confound mitochondrial heteroplasmy analysis will vary between modalities and approaches to library preparation. In their ATAC study, Lareau et al. addressed the issue of NUMTs by masking nuclear genomic regions which in silico mitochondrial derived reads mapped to. As a result all relevant reads would map to the mitochondrial genome. The high copy number of mtDNA in each cell relative to nuclear sequences should result in a low error rate of NUMTs being assigned to the mitochondrial genome. Others have found that by aligning exclusively to the mitochondrial genome scRNA achieves superior breadth coverage of the mitochodnrial genome in mice, which possess NUMTs that affect 6 mtDNA genes [63].

### 5.7. High-Throughput Droplet Data

High-throughput droplet-based approaches to single-cell sequencing are the dominant modality with 10× chromium being one of the leading methods of library preparation available. Such short-read approaches are biased with reads being sequenced from the 3${}^{\prime }$ or 5${}^{\prime }$ end of a transcript preventing consistent full mitochondrial coverage per cell. Recent work, however, has shown that in spite of this end bias, 10× 3${}^{\prime }$ and 5${}^{\prime }$ libraries have low levels of coverage up to 10 Kbp from the 3${}^{\prime }$ and 5${}^{\prime }$ ends of the transcripts respectively [87]. Under certain circumstances, these regions could allow for high heteroplasmy variants to be called as they have a high detection rate even at low depth of coverage, and are less likely to be confused with errors. Whilst such coverage would be heterogeneous on per cell level, the large throughput of these techniques implies that even if a small proportion of cells within a dataset yield coverage at points far from the transcript end, enough cells covering identical regions could still enable discovery of heterogeneity. Regardless of this, high-throughput approaches yield consistent coverage near transcript ends, which should enable reliable variant detection in those regions of the genome.

## 6. Conclusions

Studying heteroplasmy at the granular level of the single cell has important applications from lineage tracing cells in clonal populations to potentially assessing the pathological impact of such mutations on gene expression. Here, we surveyed several recent approaches in the literature and their application. Subsequently, we highlighted several technical limitations of current technology, both in terms of experimental approach such as read depth, ambient RNA and end bias, and potential software induced errors linked to quality control filters and NUMTs. Whilst such technical issues can confound analyses, biological heterogeneity can and has been detected with such approaches. As sequencing technology advances, finer levels of granularity will be observable, and more robustly. However, given the volume of data already published covering many different tissue types and the successful explorations of mitochondrial variation using these approaches, we believe that current data could be further leveraged to explore mitochondrial heterogeneity at the level of the single cell.

## Figures and Tables

**Figure 1 biology-10-00503-f001:**
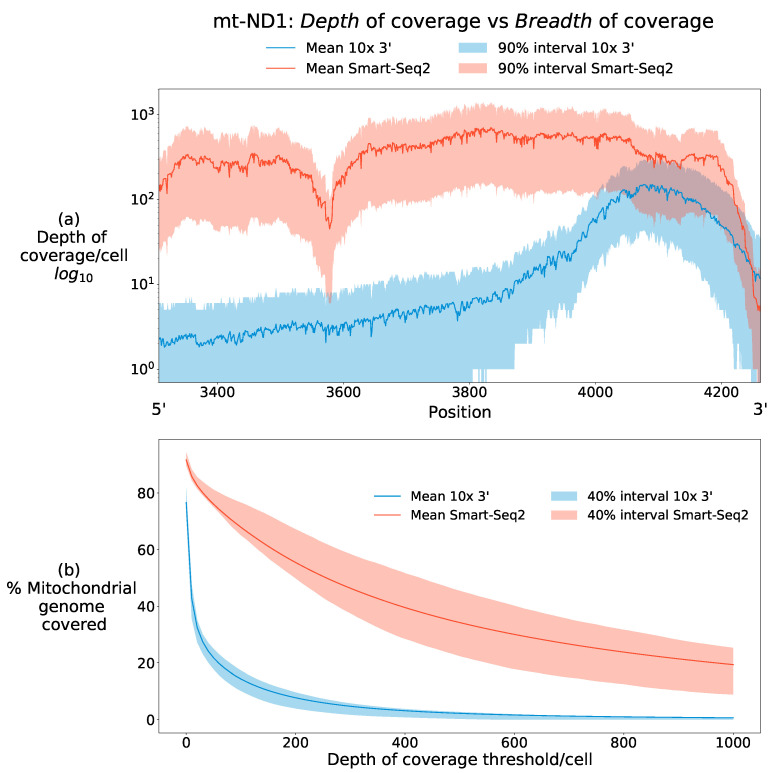
Breadth of coverage and depth of coverage differ between sequencing types. (**a**) Here, both meanings of coverage are illustrated for the mt-ND1 gene from two scRNA-seq data types [31,32]. It can be seen that 10× 3${}^{\prime }$ data (blue) has a high mean depth of coverage at the 3${}^{\prime }$ end of the transcript, with many sequenced reads aligning to this position. The mean depth of coverage rapidly declines at bases with positions roughly <4000 moving in the 5${}^{\prime }$ direction, behaviour typical of 10× 3${}^{\prime }$ data (see Section 3.1). 10× data is therefore said to have a narrow *breadth* of coverage of the mt-ND1 transcript. Smart-Seq2 (red) in contrast has a higher mean depth of coverage across almost the entire length of the transcript and can therefore be described as having both high depth and breadth of coverage. Central 90% quantiles demonstrate the heterogeneity in both coverage measures across cells, with some cells possessing far less depth of coverage in certain regions, posing a challenge for robustly identifying mutations across the mitochondrial genome in all cells. (**b**) Smart-seq2 shows superior breadth of coverage of the mitochondrial genome at all depth thresholds. This difference reflects the 3${}^{\prime }$ bias of 10× scRNA sequencing data, limiting its breadth of coverage at modest depth thresholds, where Smart-Seq2 maintains much more stable breadth of coverage up to high depth thresholds. Both methodologies, however, show reductions in the proportion of cells attaining large breadth of coverage as depth thresholds increase.

**Figure 2 biology-10-00503-f002:**
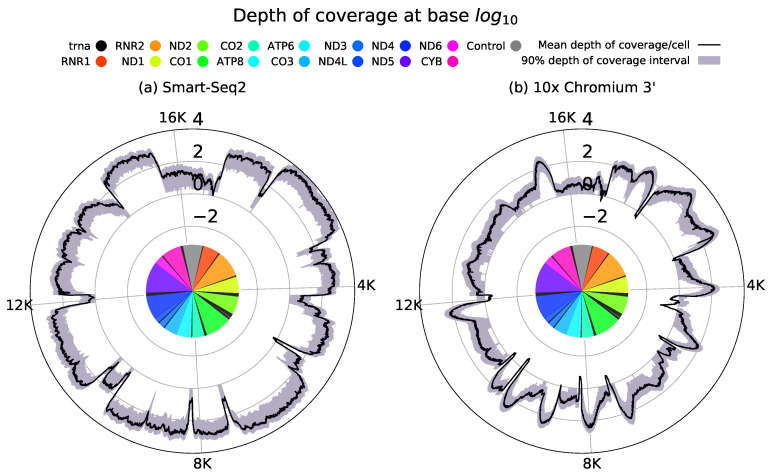
Sequencing protocols demonstrate a high degree of variation in both depth and breadth of coverage they attain across the mitochondrial genome. (**a**) Full-length Smart-Seq2 data [32] demonstrates broadly uniform coverage across transcript lengths. 10× 3${}^{\prime }$ protocols achieve a far higher throughput of cells, than full-length approaches. (**b**) However, 10× 3${}^{\prime }$ data [31] has a poorer breadth of coverage of the mitochondrial genome due to reads preferentially capturing the 3${}^{\prime }$ poly A tails of transcripts. This results in the pronounced depth of coverage which can be observed in the 3${}^{\prime }$ end of each gene. High throughput combined with consistent 3${}^{\prime }$ coverage could enable 3${}^{\prime }$ data to facilitate the discovery of mitochondrial heterogeneity. Such coverage plots can be generated by counting the reads aligned to each position in the mitochondrial genome in each individual cell.

## Data Availability

The scRNA-seq data for Figure 1 and Figure 2 were obtained from GEO with accession numbers GSE135922 and GSE81547 for the 10× 3${}^{\prime }$ and Smart-Seq2 datasets respectively.

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
