# Peer review of "Discovering Cellular Mitochondrial Heteroplasmy Heterogeneity with Single Cell RNA and ATAC Sequencing"

_biology, 2021, doi:10.3390/biology10060503_

Round 1

Reviewer 1 Report

This is a useful, timely and comprehensive review of an elegant approach to heteroplasmy measurements in mtDNA. As someone who for most of his career advocated single cell analysis of heteroplasmy (see one of the references below), I am glad to see that this approach is being used in a much more efficient way with modern sequencing methods/single cell omics.

I think one important point to emphasize more is that single cell heteroplasmy measurements are  inherently more error resistant than “homogenate” heteroplasmy measurements , because, in a sense, measurements in different cells can serve as cross-controls to each other. This allows to measure with high confidence mutations that in homogenate have no chance to be detected, this applies in particular to ‘small clonal expansions’ such as late (I mean late in life) somatic mutations in nonproliferating cells (neurons or musle fibers), where a mutation present in a given single cell )and perhaps in no other cell) at perfectly measurable fraction in homogenate will have a hopelessly low fraction. Same applies to local mutations in proliferating cells such as clonal expansions in colonic crypts.

Minor comment: please check bibliography: I detected at least one reference mismatch, but there may be more: line 530: Lareau, et al is not [63]

Reference:

Clonally expanded mtDNA point mutations are abundant in individual cells of human tissues Ekaterina Nekhaeva, Natalya D Bodyak, Yevgenya Kraytsberg, Sean B McGrath, Nathalie J Van Orsouw, Anna Pluzhnikov, Jeanne Y Wei, Jan Vijg, Konstantin Khrapko. Proceedings of the National Academy of Sciences 99 (8), 5521-5526 2002

Author Response

We welcome the reviewer’s comments and feedback. We agree with the reviewer’s comments and have incorporated this into the manuscript with the relevant citation around lines 67-71.

We further appreciate the detection of the citation mismatch and have made the relevant amendments. We detected at least one further such error in our manuscript and thank the reviewer for drawing our attention to this.

We appreciate the time taken to review our manuscript and believe your input has helped us to improve it.

Reviewer 2 Report

Dear author and editor, 

thanks for the opportunity. These review brings new and important analysis on new approaches on mtDNA study, well writing and with great contribution to the field. 

I have some suggestions bellow: 

Line 16: add space before “scATAC-seq”

Line 21: this is true for animals, but not for plants for example, please rephrase ate “unlike other organelles”

Line 23: What is expected is homoplasmy, please rephrase to a better reading.

Line 41: Beyond the putative errors during genome assembly!

Line 58: remove the highlight text

Line 77: add reference of this first study!

Line 100: would better the term “the variation” instead “heterogeneity”

Figure 1 caption: I would recommend using (a) and (b), adding to the figure the notation.

From page 6: Check headings and subheadings

Line 185: revise the title 3`and 5` approaches for a clearer subtitle

Line 203: Consider revise, In such what? I know it is UMI, but leave as clear as possible for readers  

Line 254: Remove the highlighted term.

Lines 254 – 263: I was a bit confused in this paragraph.

Figure 2, how these maps were generated?

Lines 385 and 398 and all other cases: Change the starting of the sentence, not with the citation.

Lines 561: individual cellular lineage? Or in an evolutionary context?

Line 626: References??? Other eukaryote species? (example: https://doi.org/10.1016/j.crvi.2013.11.007)

Author Response

We thank the reviewer for their feedback and believe that these suggestions improve the clarity of the manuscript. We address each point below

Reviewer 2 response to each comment

I have some suggestions bellow: 

Line 16: add space before “scATAC-seq”

Corrected

Line 21: this is true for animals, but not for plants for example, please rephrase ate “unlike other organelles”

We agree and thank the reviewer for highlighting this

Line 23: What is expected is homoplasmy, please rephrase to a better reading.

We disagree with this comment. The reference, at the end of the particular sentence you highlight shows that all humans do in fact harbour heteroplasmic mutations which are not necessarily homoplasmic. I believe the sentence in our manuscript accurately reflects this. Further elaboration would allow us to understand the source of confusion.

Line 41: Beyond the putative errors during genome assembly!

The authors agree that errors during genome assembly, as well as sequencing, place limitations on bulk analytic approaches. However, the emphasis of this sentence intended to be the limitations of the resolution of these approaches, ie the ability to explore variation at the coarse level of individual/tissue level, vs the cellular level. I have amended the sentence to further emphasise its intended meaning.

Line 58: remove the highlight text

There does not appear to be highlighted text on line 58. Please reiterate to help understanding.

Line 77: add reference of this first study!

Amended

Line 100: would better the term “the variation” instead “heterogeneity”

For the purposes of consistency throughout the manuscript, we maintain the use of the term heterogeneity.

Figure 1 caption: I would recommend using (a) and (b), adding to the figure the notation.

We agree and have adjusted the figure captions in both figures 1 and 2, as well as the manuscript to address this.

From page 6: Check headings and subheadings

Line 185: revise the title 3`and 5` approaches for a clearer subtitle

A clear subtitle has be given.

Line 203: Consider revise, In such what? I know it is UMI, but leave as clear as possible for readers  

We have revised this to bring greater clarity.

Line 254: Remove the highlighted term.

Lines 254 – 263: I was a bit confused in this paragraph.

For both of the above points, based on your suggestions, we have rephrased the entire paragraph to make for clearer reading.

Figure 2, how these maps were generated?

A succinct description was added to figure 2.

Lines 385 and 398 and all other cases: Change the starting of the sentence, not with the citation.

We agree that the previous phrasing was inappropriate and have altered the sentence structure to reflect this

Lines 561: individual cellular lineage? Or in an evolutionary context?

We have corrected this ambiguity and thank the reviewer for highlighting it.

Line 626: References??? Other eukaryote species? (example: https://doi.org/10.1016/j.crvi.2013.11.007)

We have taken your recommendation here on board, including the citation as well as another which supports this claim in the manuscript.